# Evaluating the Vertical Accuracy of DEM Generated from ZiYuan-3 Stereo Images in Understanding the Tectonic Morphology of the Qianhe Basin, China

**Zhiheng Liu** [1,†] **, Ling Han** [2,3,*,†] **, Zhaohui Yang** [1] **, Hongye Cao** [1] **, Fengcheng Guo** [4] **, Jianhua Guo** [5] **and Yiqi Ji** [1]

1  College of Geological Engineering and Geomatics, Chang'an University, Xi'an 710064, China; liuzhiheng@chd.edu.cn (Z.L.); 2018126045@chd.edu.cn (Z.Y.); caohy@chd.edu.cn (H.C.); yiqiji12@chd.edu.cn (Y.J.)
2  College of Land Engineering, Chang'an University, Xi'an 710064, China
3  Shaanxi Key Laboratory of Land Consolidation and Rehabilitation, Xi'an 710064, China
4  School of Geography, Geomatics and Planning, Jiangsu Normal University, Xuzhou 221116, China; fchguo@jsnu.edu.cn
5  School of Electrical and Information Engineering, Tianjin University, Tianjin 300072, China; g_j_h@tju.edu.cn
*  Correspondence: hanling@chd.edu.cn; Tel.: +86-130-0841-8178
†  These authors contributed equally to this work.

**Abstract:** Currently available high-resolution digital elevation model (DEM) is not particularly useful to geologists for understanding the long-term changes in fluvial landforms induced by tectonic uplift, although DEMs that are generated from satellite stereo images such as the ZiYuan-3 (ZY3) satellite include characteristics with significant coverage and rapid acquisition. Since an ongoing analysis of fluvial systems is lacking, the ZY3 DEM was generated from block adjustment to describe the mountainous area of the Qianhe Basin that have been induced by tectonic uplift. Moreover, we evaluated the overall elevation difference in ZY3 DEM, Shuttle Radar Topography Mission ($1'' \times 1''$) (SRTM1), and Reflection Radiometer Global Digital Elevation Model (ASTER GDEM) by using the Ice Cloud and Land Elevation Satellite/Geoscience Laser Altimeter (ICESat/GLAH14) point cloud and a DEM of 1:50,000 scale. The values of the root mean square error (RMSE) of the elevation difference for ZY3 DEM were 9.31 and 9.71 m, respectively, and are in good agreement with SRTM1. The river long profiles and terrace heights were also extracted to compare the differences in channel steepness and the incision rates with SRTM1 and ASTER GDEM. Our results prove that ZY3 DEM would be a good alternative to SRTM1 in achieving the 1:50,000 scale for DEM products in China, while ASTER GDEM is unsuitable for extracting river longitudinal profiles. In addition, the northern and southern river incision rates were estimated using the ages and heights of river terraces, demonstrating a range from 0.12–0.45 to 0.10–0.33 m/kyr, respectively. Collectively, these findings suggest that ZY3 DEM is capable of estimating tectonic geomorphological features and has the potential for analyzing the continuous evolutionary response of a landscape to changes in climate and tectonics.

**Keywords:** Qianhe Basin; ZY3 stereo imagery; DEM; active tectonic

## 1. Introduction

Changes in elevation that are produced by digital elevation model (DEM) can reflect surface processes that result from tectonic uplift and provide details about the displacement of a fault [1–4]. Increasing the resolution of DEMs, therefore, implies that more surface geological phenomena and geometries such as landslides [5–7], seismic processes [8], active faulting [9,10], volcanic geomorphology [11], rock damage [12], and the extent of glaciation [13,14] can be interpreted. Several studies have focused on quantifying the response of a transient landscape to active faulting based on DEMs [3,9,15], replacing the requirement for time-consuming and laborious geological fieldwork to track neotectonic movements and landform evolution [16,17].

Traditional methods have focused on digitizing the contour lines of topographic maps or real-time dynamic GPS surveys to generate high-resolution DEMs. However, both topography and position limit the accuracy of such measurements. Although light detection and ranging (LiDAR) and synthetic aperture radar (SAR) technologies allow fault ruptures to be captured at the decimeter level [18,19], the high cost of high coverage DEMs limits the development of high-resolution geological measurements, and therefore cannot supersede global sets [3]. The recent development of unmanned aerial vehicles (UAVs) that use structure-from-motion (SfM) [20–22] has led to the possibility of producing more efficient DEMs for tracking fault deformation. However, results depend heavily on climate and geographic conditions and the use of this technology to cover large areas is expensive. With the advent of digital photogrammetry, DEMs generated from stereo satellite/aerial images [23,24] have become a way to record the evolution of tectonic landforms [4,7,25]. In particular, the larger coverage provided by these images and the ease of availability means that DEMs produced using this method could provide more detailed information concerning the rate at which spatial and temporal changes are taking place in the tectonic uplift and are helpful in describing mountainous areas with higher slopes that are difficult to access physically, such as the Qianhe Basin on the southwest margin of the Ordos Block in China [9].

Several global high-resolution DEMs are available for active tectonic areas, meaning that surface displacements and differences in elevation can be estimated to quantify the slip rate from the fluvial geomorphology [15,26]. Although comparisons of existing data sets such as the Shuttle Radar Topography Mission (SRTM) and the Advanced Spaceborne Thermal Emission and Reflection Radiometer Global Digital Elevation Model (ASTER GDEM) have been made for higher mountainous areas for quantifying the response of longitudinal river profiles [3], few studies have focused on the accuracy and efficiency of DEMs, in particular, those produced from stereo satellite images. River terraces record uplift information, and an analysis of the incision rates is useful in understanding these formations [27–29]; however, no such study has been conducted in Qianhe Basin. A multitemporal stereo pair can track a long-term sequence slip rate that no other DEM can delineate. Therefore, we compared high-resolution DEM, extracted from the Chinese ZY3 stereo satellite images, with results from SRTM and ASTER GDEM to understand the fluvial geomorphic evolution of the natural active faulting region of Qianhe Basin [2,9,30] in order to analyze the river longitudinal profiles and terraces in the area.

## 2. Study Area and Datasets

### 2.1. Study Area

Over recent years, many qualitative and quantitative studies have discussed the uplift rate of the active faulting in the Weihe Basin and the surrounding area using geomorphic indices extracted from DEMs [2,31–36]. As one of the main tributaries of the Weihe River, the Qianhe River is located at the joint zone between the Liupan–Longxi mountains and the southwest margin of the Ordos Block (Figure 1) [9,35], with a catchment area of 3940.1 km². Qianhe Basin was formed via the ongoing uplift of four active faults (Figure 1), but it remains difficult to estimate slip rates because of the lack of geographic access. Zhang, Fan, Chen, Li and Lu [2], therefore, extracted the hypsometric integral (HI), stream length gradient (SL), drainage basin asymmetry (AF), and drainage basin shape (BS) from ASTER GDEM to assess the tectonic activity in this area, and found that the uplift area of the Liupan–Longxi mountains was induced by the Taoyuan–Guichuansi Fault (TGF) and the Guguan–Guozhen Fault (GGF), while the southwest Ordos Block was tilted by the left-lateral Qishan–Mazhao Fault (QMF) (Figure 1). Liu, et al. [9] extracted the normalized channel steepness ($k_{sn}$) and knickpoint retreat rates (0.3–27.3 mm/year) from the 24 main tributaries of the Qianhe River in SRTM to investigate the fault uplift, which indicated that the incisional pattern of the drainage network in the Qianhe Basin was mainly formed by active faulting [9]. However, most previous studies have ignored the accuracy and effectiveness of datasets when extracting geomorphological parameters. High vegetation

coverage can block the radar in active tectonic zones, resulting in voids in the DEMs [37] and incorrect indices, which lead to the faults not being constrained. The river terrace sequences that have been apparent in the Qianhe Basin since the late Cenozoic have also been linked to neotectonic movement and the geomorphic response time [35]. The area is characterized by a semi-arid and semi-humid climate [2], with annual precipitation ranging from 653.4 to 924.3 mm [38]. In summary, the Qianhe Basin is an ideal site for comparing and validating different DEMs for analyzing long river profiles and incisions in the tectonic geomorphology.

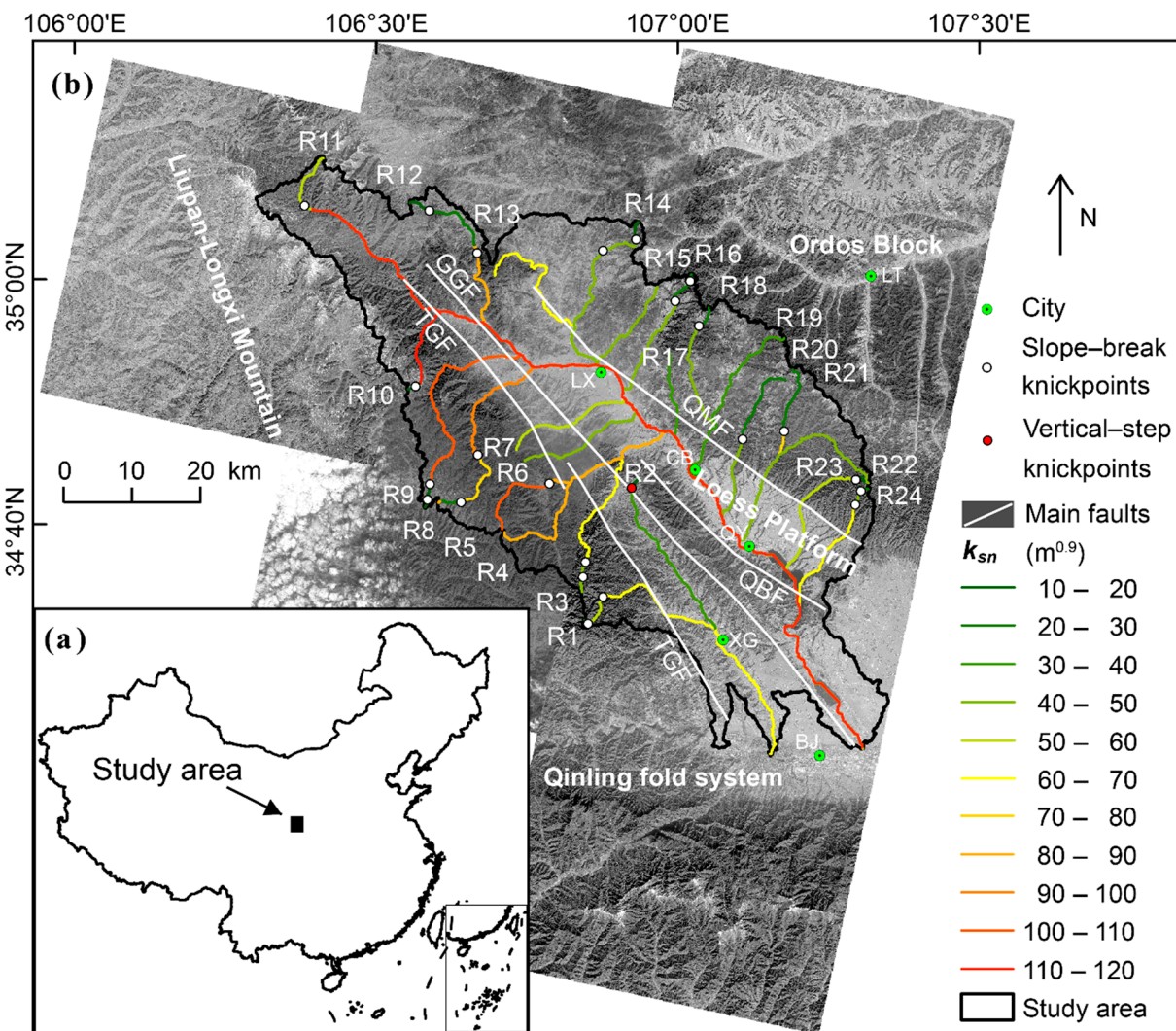

**Figure 1.** (**a**) Map of China; (**b**) The study area and ZiYuan-3 (ZY3) stereo satellite images coverage region. The ZY3 stereo satellite images are collected from http://www.satimage.cn, accessed on 18 March 2021. The white lines are the main faults in the Qianhe Basin, i.e., QMF, Qishan–Mazhao Fault; QBF, Qianyang–Biaojiao Fault; GGF, Guguan–Guozhen Fault; TGF, Taoyuan–Guichuansi Fault. The main cities include LX, Longxian; CB, Caobi; QY, Qianyang; XG, Xiangong; BJ, Baoji; LT, Liangting. Twenty-four tributaries and knickpoints in the Qianhe Basin are collected from [9].

### 2.2. Datasets

#### 2.2.1. ZiYuan-3 (ZY3) Satellite Image

In mountainous areas, traditional global DEM datasets such as SRTM and ASTER GDEM cannot be updated regularly because of the high costs involved and difficulties with political access. The advent of stereo satellite/aerial images has provided a new means of generating DEMs for surveying and mapping, rendering it possible to quantify the response of river longitudinal profiles to tectonic uplift over tens to hundreds of years,

especially tracing the changes in slip rate of the faults, for 100 years before and after. ZY3, the first Chinese civilian stereo mapping satellite [23,39,40], was launched on 9 January 2012, followed by the ZY3-02 satellite that was placed in orbit on 30 May 2016. The sun-synchronous orbit reaches a height of 506 km, with a revisiting period of 5 days (Table 1), and therefore can seamlessly cover several regions, ranging over 84° of the north–south latitude of the earth. The ZY3 sensor is composed of three charge-coupled device (CCD) linear pushbroom panchromatic cameras (i.e., forward (FWD), backward (BWD), and nadir (NAD)) and one multispectral camera (MUL) [23]. The NAD camera has the highest resolution as compared with the FWD, BWD, and MUL cameras (Table 1), with a ground sample distance of 2.1 m, and covers $51 \times 51$ km² [41]. Datasets from the ZY3 images can be applied in topographic mapping at a scale of 1:50,000. Six stereo pairs of ZY3 panchromatic NAD and FWD images, with an overlap of <5% cloud, were used to investigate the Qianhe Basin, as shown in Figure 1 and Table 2. No seismic hazards have been reported in the Qianhe Basin since the ZY3 launched.

**Table 1.** Main parameters of the ZY3 satellite, collected from http://www.satimage.cn, accessed on 18 March 2021.

| Sensors | Band Order | Wavelength (μm) | Spatial Resolution (m) | Ground Swath (km) | Side Rotation | Revisit Period (d) |
|---|---|---|---|---|---|---|
| Forward image | – | 0.50~0.80 | | | | |
| Backward image | – | 0.50~0.80 | 3.5 | 52 | | 3~5 |
| Nadir image | – | 0.50~0.80 | 2.1 | 51 | | |
| | 1 | 0.45~0.52 | | | ±32° | |
| Multiple image | 2 | 0.52~0.59 | | | | |
| | 3 | 0.63~0.69 | 5.8 | 51 | | 5 |
| | 4 | 0.77~0.89 | | | | |

**Table 2.** Metadata of the ZY3 stereo images of the Qianhe Basin.

| No | Filename | Row | Path | Acquisition Time | Solar Zenith | Solar Azimuth | Satellite Zenith | Satellite Azimuth | Cloud Coverage |
|---|---|---|---|---|---|---|---|---|---|
| 1 | L1A0001497551 | 137 | 23 | 2013-12-11 11:52:12 | 30.5° | 165.1° | 84.7° | 101.5° | 0% |
| 2 | L1A0001051630 | 137 | 22 | 2013-02-24 11:50:36 | 42.1° | 154.4° | 88.6° | 268.6° | 0% |
| 3 | L1A0001051631 | 138 | 22 | 2013-02-24 11:50:42 | 42.4° | 154.2° | 88.6° | 268.6° | 4% |
| 4 | L1A0000931653 | 137 | 21 | 2013-01-01 11:47:12 | 29.8° | 162.1° | 89.7° | 206.1° | 0% |
| 5 | L1A0000931654 | 138 | 21 | 2013-01-01 11:47:15 | 30.4° | 161.9° | 89.7° | 206.2° | 0% |
| 6 | L1A0000931655 | 139 | 21 | 2013-01-01 11:47:24 | 30.5° | 161.8° | 89.7° | 206.3° | 0% |

2.2.2. Shuttle Radar Topography Mission Digital Elevation Model (SRTM DEM) and Reflection Radiometer Global Digital Elevation Model (ATSER GDEM)

To assess the accuracy of the DEM generated from ZY3 stereo images, the freely accessible global DEM datasets on the United States Geological Survey (USGS) Earth Explorer interface (https://earthdata.nasa.gov/, accessed on 18 March 2021), such as SRTM and ASTER GDEM, were selected for comparison of the study area. SRTM DEM, which scans the 60° N to 56° S regions of the Earth from a radar topographic measurement implemented jointly by the National Aeronautics and Space Administration (NASA) and National Imagery and Mapping Agency (NIMA), USA, in 2000, covering >80% of the surface area of the Earth [37,42,43]. Owing to the differential interferometry, the SRTM DEM was generated from C-band synthetic aperture radar (SAR) data [44] at centimeter-level accuracy. The SRTM DEM is divided into two main data types, i.e., SRTM1 (1″ × 1″) and SRTM3 (3″ × 3″), with resolutions of 30 m and 90 m, respectively. The validation for the SRTM mainly includes the following: (a) dynamic differential GPS, (b) deployment of angular reflectors with high reflectivity and precise coordinates for clear imaging on radar images, and (c) radar data in the sea surface area. Previous studies using the reference DEM generated from the high-accuracy laser point cloud data describing the Loess Plateau on the North China Plain of China have shown that the vertical accuracy of SRTM is better than 8.8 m in China [45,46], which is related to the presence of voids in the SRTM [3].

On the basis of optical stereo photogrammetry, the ASTER GDEM was produced by the Ministry of Economy, Trade and Industry (METI) of Japan and the United States National Aeronautics and Space Administration (NASA), generated from 1.5 million near-infrared images obtained from the TERRA satellite launch in June 2009 [47], covering the area from 83° N to 83° S (99% of the Earth's surface). Due to the improved position of the satellite ephemeris and the sensor model of the ASTER instrument, the points in ASTER DEM can be fully and precisely located (with absolute coordinates) without the need for external control points. Each image comprises 4200 rows × 4100 columns, corresponding to approximately 60 × 60 km on the surface. Then, each image can generate some small DEMs and the final ASTER GDEM is obtained by averaging these DEMs' elevation and correcting residual anomalies. The ASTER GDEM data were provided at a spatial resolution of 1 arc sec (~30 m), referenced horizontally to WGS84, and vertically to EGM96 [48]. The ASTER GDEM validation team [49] has indicated that the vertical accuracy of global ASTER GDEM is approximately 20 m at 95% confidence. Specifically, the ASTER GDEM has the following two types of validation: (a) for METI, ASTER GDEM vertical accuracy was mainly focused on the comparisons with two high-resolution DEMs (GSI 5 m and GSI 10 m) [49] and (b) for NASA, the validation was implemented by using corresponding 1 arc-second elevation data from the USGS National Elevation Dataset (NED) [49]. Previous studies have also assessed the vertical accuracy of ASTER GDEM for the southeast margin of the Chinese Loess Plateau, with an accuracy of 12.6 m [46]. The main difference between these two DEMs is that SRTM radars can penetrate vegetation canopies [37]. This comparison will help make more informed decisions for landscape evolution analyses in active tectonic areas.

## 3. Method

### 3.1. ZY3 DEM Generation

The traditional method of DEM generation from stereo pairs is mainly based on the binocular vision matching proposed by Marr and Nishihara [50], while the DEM generation transforms the two cameras as the human eye to scan the same surface. More specifically, the surface is scanned continuously via the cameras on the satellite to obtain overlapping images (Figure 2), and the true three-dimensional coordinates can be obtained via the collinear equation (Equations (1) and (2)), in which the relationship between the object point (A), image point ($a_1$), and projection center ($S_1$) describe a line (Figure 2).

$$x = -f\frac{a_1(X - X_s) + b_1(Y - Y_s) + c_1(Z - Z_s)}{a_3(X - X_s) + b_3(Y - Y_s) + c_3(Z - Z_s)}, \tag{1}$$

$$y = -f\frac{a_2(X - X_s) + b_2(Y - Y_s) + c_2(Z - Z_s)}{a_3(X - X_s) + b_3(Y - Y_s) + c_3(Z - Z_s)}, \tag{2}$$

where $(x, y)$ is the image point corresponding to the object point $(X, Y, Z)$; $(X_s, Y_s, Z_s)$ is the projection center; $f$ is the focal distance; and $a_i$, $b_i$, and $c_i$ ($i = 1, 2, 3$) are the perimeters generated by the exterior orientation elements ($\varphi$, $\omega$, and $\kappa$), respectively. $(x_0, y_0)$ is the center of the image.

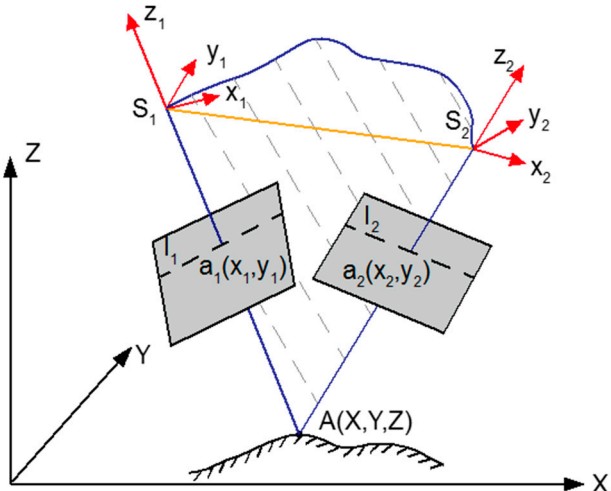

**Figure 2.** Digital elevation model (DEM) generation from stereo images, modified from [51]. $a_1$ and $a_2$ are the image points corresponding to the object point (A) on the true ground, $S_1$ and $S_2$ are the projection center, respectively. $\overrightarrow{S_1S_2}$ is the photographic baseline, and the shadow lines are shown as the epipolar plane between the photographic baseline and the object point. The intersection line (e.g., $l_1$ and $l_2$) between the epipolar plane and the image plane is the epipolar line.

The traditional way above cannot satisfy the requirements for the push-broom scanning system (i.e., ZY3 satellite), as the inner ($f$, $x_0$, $y_0$) and exterior orientation elements of the ZY3 satellite are not accessible to users, but instead provide a Rational Function Model (RFM) file that represents the relationship between a normalized image point ($l$, $s$) and the corresponding object point in object space (*Lat*, *Lon*, and *Hei*) [52,53], which can be expressed as follows:

$$l = \frac{Num_l(Lat, Lon, Hei)}{Den_l(Lat, Lon, Hei)}, \tag{3}$$

$$s = \frac{Num_s(Lat, Lon, Hei)}{Den_s(Lat, Lon, Hei)}, \tag{4}$$

where (*Lat*, *Lon*, and *Hei*) are the normalized latitude, longitude, and height in object space that are found using the following equations:

$$Lat = \frac{lat - lat_{offset}}{lat_{scale}}, \tag{5}$$

$$Lon = \frac{lon - lon_{offset}}{lon_{scale}}, \tag{6}$$

$$Hei = \frac{hei - hei_{offset}}{hei_{scale}}, \tag{7}$$

where $lat_{\text{offset}}$, $lon_{\text{offset}}$, and $hei_{\text{offset}}$ are the offset coefficients of the latitude, longitude, and height, respectively. $lat_{\text{scale}}$, $lon_{\text{scale}}$, and $hei_{\text{scale}}$ are the scale coefficients of the latitude, longitude, and height, respectively. $Num_l$ (*Lat*, *Lon*, *Hei*), $Den_l$ (*Lat*, *Lon*, *Hei*), $Num_s$ (*Lat*, *Lon*, *Hei*), and $Den_s$ (*Lat*, *Lon*, *Hei*) are the three rational polynomials of the object point (*Lat*, *Lon*, and *Hei*), and can be described as follows:

$$\begin{aligned} F(B, L, H) = \ & a_1 + a_2L + a_3B + a_4H + a_5LB + a_6LH \\ & + a_7BH + a_8L^2 + a_9B^2 + a_{10}H^2 + a_{11}BLH \\ & + a_{12}L^3 + a_{13}LB^2 + a_{14}LH^2 + a_{15}L^2B + a_{16}B^3 \\ & + a_{17}BH^2 + a_{18}L^2H + a_{19}B^2H + a_{20}W^3 \end{aligned}, \tag{8}$$

where $a_i$ ($i$ = 1, 2, 3, ... , 20) is the coefficient of three rational polynomials in RFM.

In general, the precision of a DEM cannot meet the requirements for system errors in the exterior orientation elements of RFM that are induced by spaceborne GPS receivers, star cameras, or gyroscopes [54]. Moreover, previous studies have shown that the horizontal and vertical RMSE is better than 15 m and is approximately 6–10 m without ground control points (GCPs) in 1:25,000 scale surveying and mapping [41,55,56]. Therefore, more GCPs are required to improve the positioning accuracy of the RFM [23]. Additionally, matching the features of the image points in the stereo images also constrains the precision during DEM generation as the stereo images cannot be aligned if the relative position of the same image points in a stereo pair cannot be matched. Therefore, to provides error correction to the coordinates of image points from RFM, six parameters ($a_0$, $a_1$, $a_2$, $b_0$, $b_1$, $b_2$) of affine transformation model are usually used to describe the block adjustment as follows:

$$line = a_0 + a_1 \cdot x + a_2 \cdot y + y \tag{9}$$

$$sample = b_0 + b_1 \cdot x + b_2 \cdot y + x \tag{10}$$

where (sample, line) are the image coordinates generated from RFM and ($x$, $y$) are the measurement coordinates of tie points (or control points) in image space.

Therefore, more GCPs (e.g., 10 to 12) are usually distributed evenly and investigated for each ZY3 stereo image (Figure 3) even though 3 points are enough.

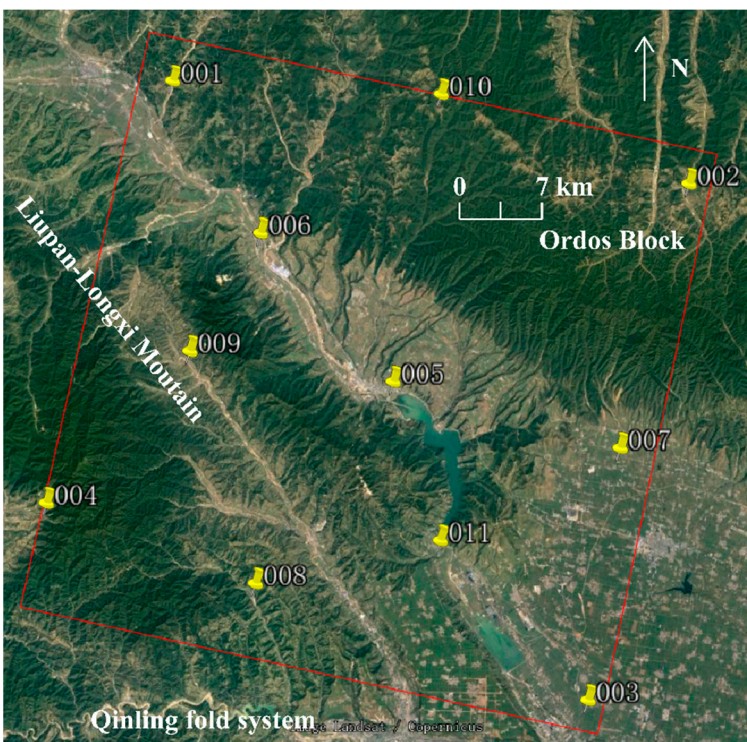

**Figure 3.** Representative distribution of ground control points (GCPs) in a single ZY3 scene region in the Qianhe Basin. The red line is the area of the ZY3 stereo images and the yellow points are the GCPs.

### 3.2. River Longitudinal Profiles

In active tectonics, river long profiles have been considered to be the key signal with which to track changes in the topography that are induced by tectonic uplift, lithological strength, climate changes, and the boundary conditions of erosion [57–60]. DEMs have been widely used in the analysis of fluvial systems and provide some empirical information for quantifying the response of a transient landscape to active faulting (e.g., [9,10,15,61–63]). The topography of the steady-state river systems in such locations (erosion ≈ uplift) is

described by a power law relationship between the local channel slope (*S*) and the upstream drainage area (*A*) (Figure 4a,b):

$$S = k_s A^{-\theta}. \tag{11}$$

where $k_s$ is the channel steepness index.

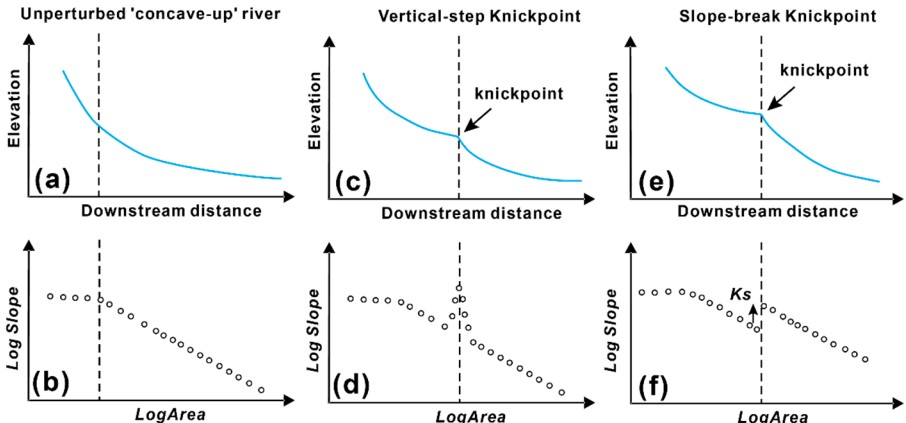

**Figure 4.** Schematic diagrams showing the river long profile knickpoint patterns and log-log slope-area plots. (**a**,**c**,**e**) The longitudinal profiles without knickpoint, with one vertical step knickpoint, and the slope break knickpoints, respectively; (**b**,**d**,**f**) The log-log slope-area plots corresponding to the longitudinal profiles, modified from [3,26,57].

Additionally, the pattern and location of the knickpoints (the point in the long river profiles where the changes in S reach a local maximum) divide the profiles into two types of steepness) (Figure 4) [64–66]. These changes have also been regarded as the main signals with which fault uplift can be investigated (see Kirby and Whipple [57], for more detailed information). More specifically, the knickpoints in the SA plot can be divided into two types, i.e., the vertical-step knickpoint (with little variation in the value of $k_{sn}$ above and below the knickpoint, Figure 4c,d) and the slope-break knickpoint (in which $k_{sn}$ increases below the knickpoint, Figure 4e,f) [57,64]. Previous studies have suggested that the slope-break knickpoint can be used to quantify the transient response to active faulting [26,57], while the vertical-step knickpoint has no tectonic significance but provides some insights into the location of faults [9].

However, this study is not concentrated on compiling or discussing the extraction of parameters in the Qianhe Basin, but rather to understand the limitations and discrepancy of the DEM datasets and evaluate the potential of using this data to investigate the contrast in river long profiles that result from the use of different DEMs.

*3.3. River Incision*

The river terrace is also a carrier that records the relative changes in tectonic activity and the climate of orogenic belts. River terraces are the result of changes in the dynamic conditions of a river, and are due to various tectonic activities, climate change, or variations in bedrock strength [27,67–70], representing the landscape after tectonic uplift and reflecting the pattern, range, and rate of tectonic deformation [28,29,70,71]. Therefore, the active tectonic zone that includes five river terraces in the Qianhe Basin has become a natural laboratory to study how fluvial systems respond to changes in both tectonics and the climate [9]. In view of this, we can extract the heights of the river terraces (H) from these DEMs by using the topography profiles to estimate the river incision rates that are induced by the tectonic uplift (Figure 5 and Equation (12)). Additionally, although there is some difference between the river undercut indicated by DEM and fieldwork, in particular, upper erosion and lower accumulation should not be ignored, and the results still

provide information for selecting the metadata for tectonic geomorphological application. Equation (12) is as follows:

$$V = \frac{H}{t}. \tag{12}$$

where the $V$ is the river incision rates for these terraces and $t$ is measured from the ESR and OSL.

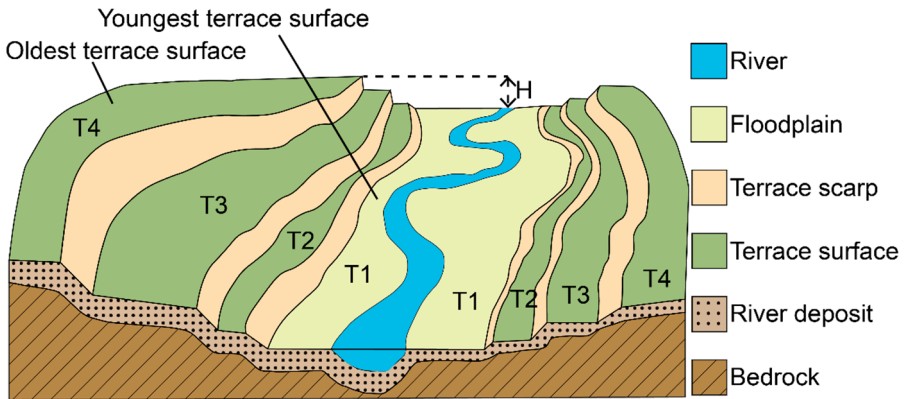

**Figure 5.** Schematic map of terrace incision rate, H is the terrace height from the river base flow level.

## 4. Results

### 4.1. DEM Vertical Accuracy Evaluation

Twenty GPS points from the field observation were used as checking points, distributed in the entire study area and their vertical error were <1 m. We also used these points to check the vertical accuracy of the three DEMs and found the RMSE values for SRTM1, ASTER GDEM, and ZY3 DEM were 5.1, 7.4, and 5.4 m in the Qianhe Graben, respectively, (see Section 4.1 for details). As shown in Figure 1 and Table 2, several clouds are present in the stereo images from the southwest margin of the Qianhe Basin. Therefore, the stereo images were first clipped as the clouds would result in large void areas in the final DEMs [72,73]. Each GCP was investigated from GPS measurements and reprojected into WGS_1984_UTM_Zone_48N with an accuracy of <1 m. Then, the ZY3 DEM for the Qianhe Basin was generated after mosaicking, projecting, and clipping, with the resulting image shown in Figure 6a. However, some visual differences remain among the three DEMs describing the Loess Platform on the southeast margin of the Qianhe Basin (Figure 6b–d). More specifically, the SRTM1 is the smoothest dataset, but overfitting has led to the loss of detailed information such as the river channel profile (Figure 6b), but this does not mean that the river networks cannot be extracted from SRTM1. Secondly, because of the high resolution and coverage of the ZY3 satellite images, the ZY3 DEM shows higher resolution and clearer river channels on the hillshade map (Figure 6c). However, more noise (e.g., roads and buildings) is present than in SRTM1, which needs to be removed from the topography map before the morphological indexes are extracted. Finally, the ASTER GDEM has the greatest amount of noise among the datasets, as indicated by the abnormal bulge that is present within the image (Figure 6e) and may increase the steepness of the channels and the height of the terraces. Moreover, the river surface is no longer flat (Figure 6f), and the lengths of the channels deviate from that of the actual channels. Therefore, more smoothing and depression-filling algorithms need to be applied to ASTER GDEM, although some geomorphological studies have utilized this model [2].

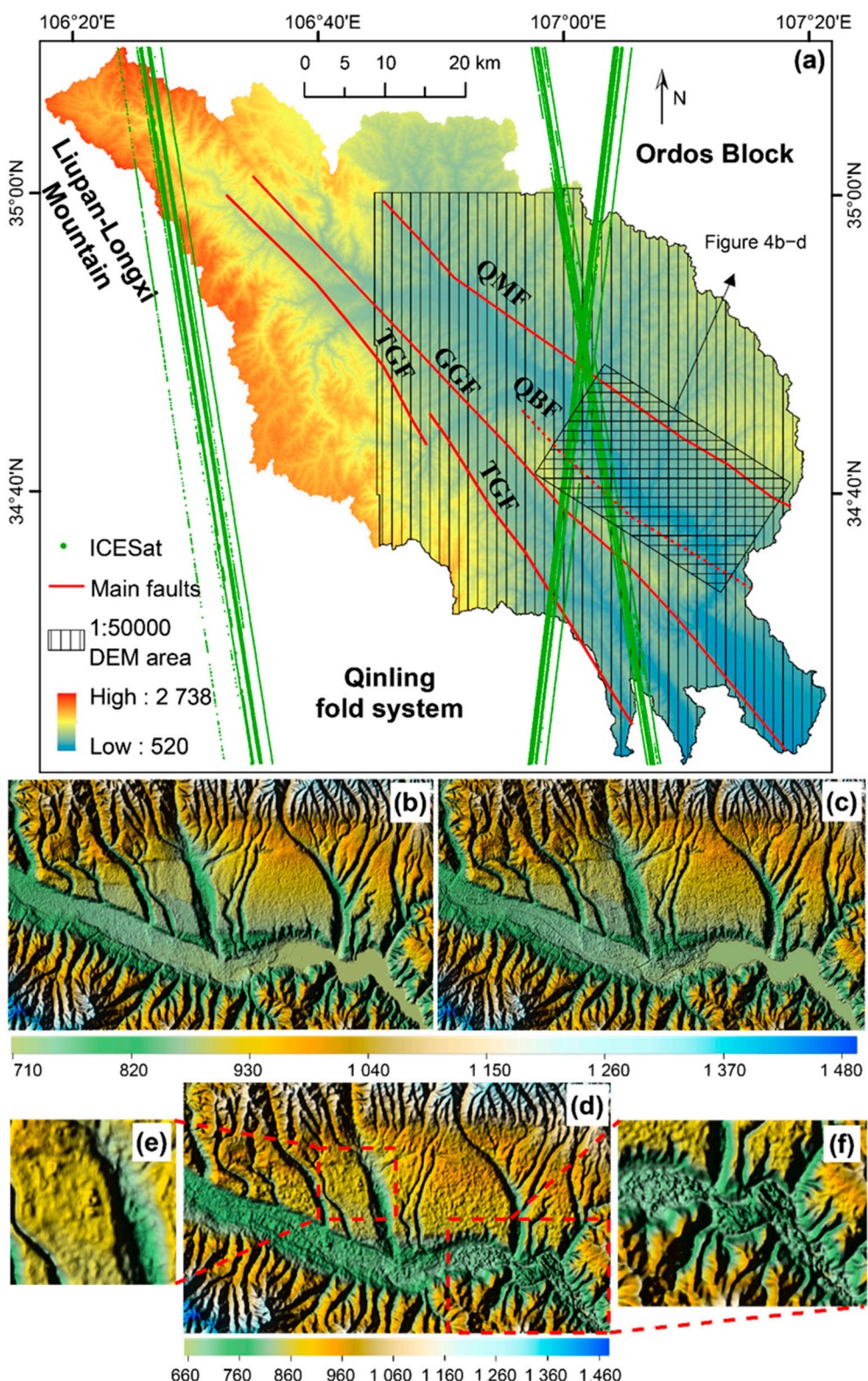

**Figure 6.** (**a**) The DEM was generated from ZY3 satellite stereo images in the Qianhe Basin. The ICESat/GLAH14 data is the elevation data from the National Snow and Ice Data Center (NSIDC, https://nsidc.org/data/icesat/, accessed on 18 March 2021). The 1:50,000 DEM data is collected from the fieldwork [74]. See Figure 1 for active faults names; (**b**), (**c**), and (**d**) are the hillshade maps of the Loess Platform for the Shuttle Radar Topography Mission (1" × 1") (SRTM1), ZY3 DEM, and Reflection Radiometer Global Digital Elevation Model (ASTER GDEM), respectively. The red dotted lines on (**e**) and (**f**) are the local magnified maps from ASTER GDEM, respectively.

Although extracting DEM from stereo images has become more applicable and widespread, the accuracy of DEM is still unpredictable and relies on the relief, vegetation coverage, sampling density of the tie points, and the distribution of GCPs and RFM [75,76]. Moreover, as the metadata describes the river long profile, the accuracy of the DEM affects the extraction of the final value of $k_{sn}$ and the knickpoints [3]. Therefore, four indexes, i.e., elevation difference (d), mean error (Mean), standard deviation (STD), and root mean square error (RMSE) (Equations (13)–(16)) [77] were used to evaluate the accuracy of ZY3 DEM with other global DEMs sets that have investigated the river long profiles of the Qianhe Basin. The equations are used as follows:

$$d = h_{DEM} - h_{ICESat/GLAH14}, \tag{13}$$

$$Mean = \frac{\sum d}{n}, \tag{14}$$

$$SD = \sqrt{\frac{\sum (d - Mean)^2}{n}}, \tag{15}$$

$$RMSE = \sqrt{\frac{\sum d^2}{n}}, \tag{16}$$

where $h_{ICESat/GLAH14}$ is the high-precision elevation data collected from the National Snow and Ice Data Center (NSIDC, https://nsidc.org/data/icesat/, accessed on 18 March 2021), with horizontal and vertical accuracies of $\pm 20$ and 18 cm [78–80].

ICESat/GLAS was launched by NASA in January 2003 and measured profiles continuously along the surface of the Earth by laser pulsation at a rate of 40 s$^{-1}$ [81]. However, the ICESat/GLAH data obtained by the NSIDC cannot be calculated directly because it uses the TOPography Experiment (TOPEX) ellipsoid, while the elevations in SRTM, ASTER GDEM, and ZY3 DEM are orthometric heights that reference the Earth Gravitational Model of 1996 (EGM96) geoid [79,82] (Figure 7).

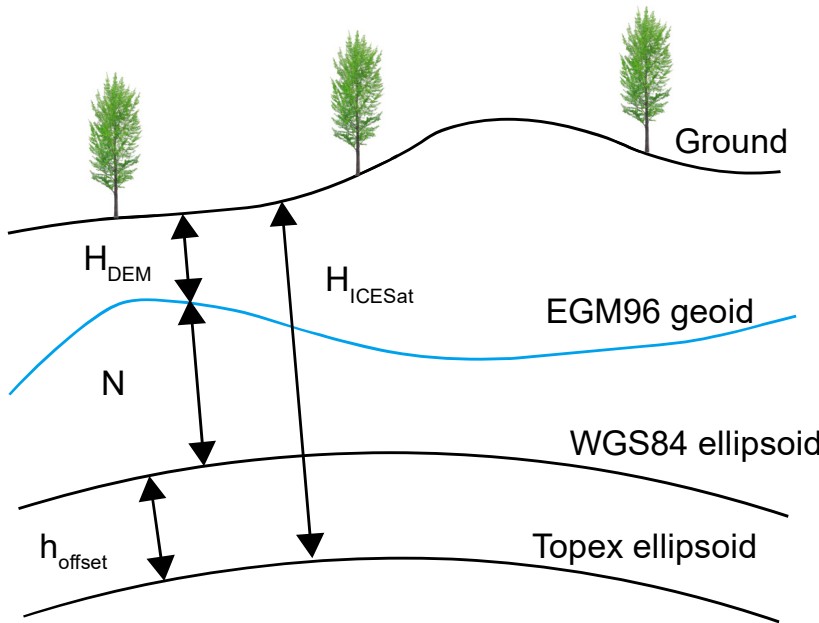

**Figure 7.** Relationship between the ICESat/GLAH data and the DEM in this study.

Therefore, 30 rails ICESat GLAH14 points (5776 in total) were collected (Figure 6a) and converted into the WGS84 ellipsoid, following Bhang et al. [83] as follows:

$$h_{WGS84} = h - N - h_{offset}. \tag{17}$$

where $h$ is the elevation obtained directly from GLAH14, $N$ is the geoid undulation, and $h_{offset}$ is a constant (~0.7 m) that represents the vertical datum difference between two ellipsoids.

Firstly, as shown in Table 3, the SD and RMSE of SRTM are the smallest of the DEMs, with values of 8.37 and 8.98 m, respectively, while the ASTER GDEM is the largest at greater than 12 m. As a result, the vertical accuracy of the SRTM1 data is higher. The mean, SD, and RMSE of the ZY3 DEM data are in the middle and close to SRTM1. All these results are consistent with several previous studies that have focused on the Ordos Block and its surroundings, such as that by Zhan et al. [84], who suggested that the RMSE of the SRTM in Shaanxi Province ranged from 3.5 to 60.7 m, decreasing from the south to the north. Guo et al. [85] indicated that the standard deviation of ASTER GDEM ranged from 7 to 14 m; Dong and Shortridge [86] estimated the root mean squared error (RMSE) of ASTER GDEM in the Changwu site of the Chinese Loess Plateau at 15.45 m for the northern Weihe Basin and eastern Qianhe Basin; Zhao, Cheng, Jiang, and Sha [45] indicated that the vertical accuracies of SRTM1, ASTER GDEM, and ZY3 DEM are approximately 8.8, 12.6, and 4.6 m, respectively, at the eastern margin of Ordos Block in Taiyuan City, and that the accuracy increased with slope.

**Table 3.** Statistics of the vertical error values for different DEMs in the Qianhe Basin.

| DEM | Points | $d_{max}$ (m) | $d_{min}$ (m) | *Mean* (m) | *SD* (m) | *RMSE* (m) |
|:---:|:---:|:---:|:---:|:---:|:---:|:---:|
| SRTM1 | | 74.07 | −49.28 | 3.24 | 8.37 | 8.98 |
| ASTER GDEM | 5776 | 91.75 | −61.28 | −3.92 | 12.32 | 12.93 |
| ZY3 DEM | | 34.27 | −35.36 | 3.15 | 8.76 | 9.31 |

Secondly, a map detailing the differences in the elevation of the Qianhe Basin in the three DEMs was drawn, and it is shown in Figure 8. The differences between the three DEMs, in terms of elevation, showed normal distribution, with variation only observed in the symmetry. The mean errors in SRTM1 and ZY3 DEM are 3.24 and 3.15 m, respectively, while the mean in ASTER GDEM is −3.92 m, indicating that most of the ASTER GDEM data collected by ICESat/GLAH14 for the Qianhe Basin show lower elevation. Moreover, the aggregation segments for these three DEMs are $-15 - 25$ m, $-30 - 20$ m, and $-15 - 25$ m, respectively, accounting for approximately 97.0%, 95.2% and 95.84% of the total. Therefore, the aggregation of SRTM1 and ZY3 DEM is better than that of ASTER GDEM, indicating that the deviation of ASTER GDEM from the true ground elevation is greater than that of SRTM1 and ZY3 DEM. As a result, more voids need to be filled before the river long profiles are extracted.

The central area in the Qianhe Basin may be ignored as the ICESat/GLAH14 points are missing. To evaluate the accuracy of the DEMs in the central area, a 1:50,000 DEM of the Qianhe Basin with a spatial resolution of 25 m and coverage of 67.2% obtained from fieldwork was considered as a reference [74] (Figure 6). With the help of the Fishnet tool in the ArcGIS software, 10,579 sample points were collected at 500 m intervals and the elevation difference in the three DEMs was calculated. As shown in Table 4, the elevation suggested by the SRTM1 data is the closest to the 1:50,000 DEM, with a standard deviation of 8.76 m. As this is close to the accuracy of SRTM1, the DEM generated from the ZY3 stereo pairs is, therefore, assumed to be appropriate as a replacement dataset for the area in which data is missing. The SD and RMSE are the largest in these DEMs, with values of 13.76 and 14.63 m, respectively. As a result, the deviation between the ASTER GDEM and the true surface is greater, indicating that the ASTER GDEM dataset is inappropriate for investigating the river long profiles. Moreover, the mean elevation difference in the ZY3 DEM is −0.98 m, suggesting that the symmetry and stability are lower than those of SRTM1.

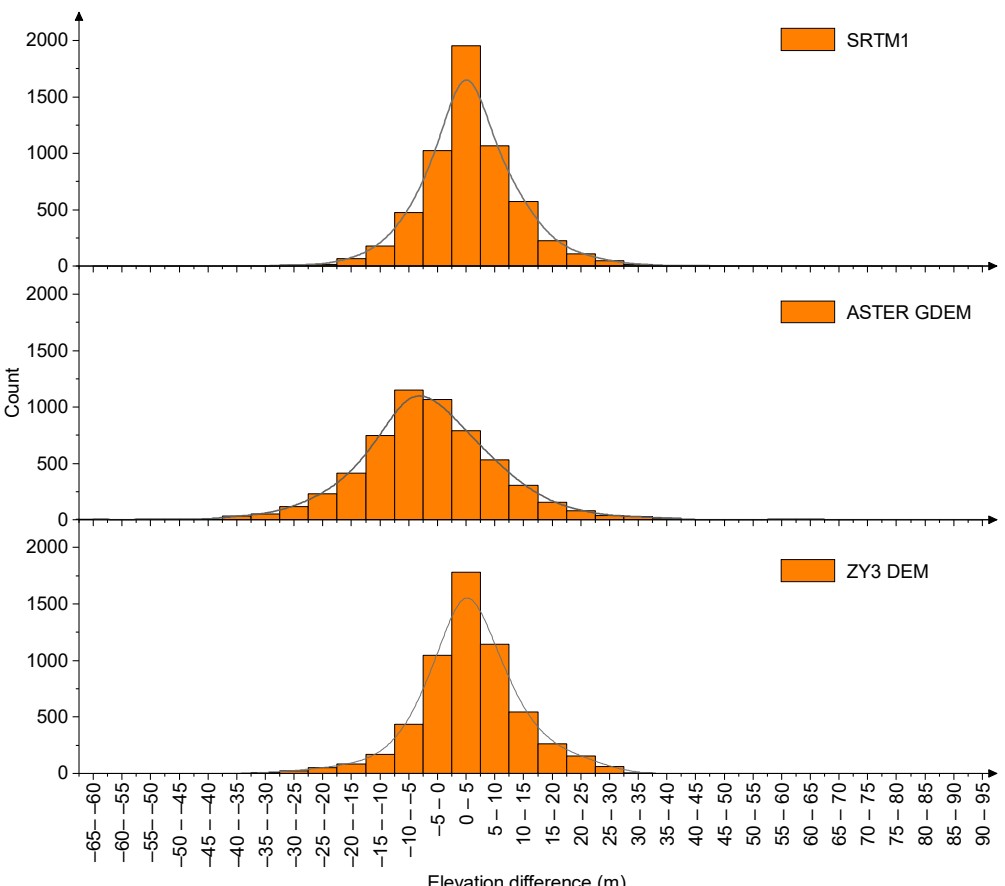

**Figure 8.** Elevation difference for these three DEMs.

**Table 4.** Statistics of the vertical error values as compared with 1:50,000 DEM.

| DEM | Points | $d_{\max}$ (m) | $d_{\min}$ (m) | *Mean* (m) | *SD* (m) | *RMSE* (m) |
|---|---|---|---|---|---|---|
| SRTM1 | | 52.40 | −56.60 | 2.23 | 8.76 | 9.04 |
| ASTER GDEM | 10,579 | 106.57 | −84.70 | −4.95 | 13.76 | 14.63 |
| ZY3 DEM | | 41.40 | −65.10 | −0.98 | 9.67 | 9.71 |

In summary, the accuracy of the SRTM1 data is higher, while the accuracy provided by ZY3 DEM is close to SRTM1, suggesting that SRTM1 and ZY3 DEM could be used as metadata to obtain a more accurate knickpoint and $k_{sn}$. However, the quality of the DEM does not mean that these datasets are necessarily optimal for applying the river hydraulic model, and further comparison and analysis is required.

### 4.2. River Longitudinal Profiles Extraction

Combining the "stream profiler" (MATLAB scripts in ArcGIS software) [87] with Equation (11), twenty-four tributaries and knickpoints that are distributed unevenly in the Qianhe Basin were extracted from SRTM1 [9] (Figure 1). To compare with the ZY3 DEM and ASTER GDEM and verify the validity and substitutability of ZY3 DEM, the $k_{sn}$ was derived from each DEM using a reference concavity $\theta_{ref}$ of 0.45 [26]. As shown in Figure 9, representative examples of the longitudinal profiles with or without knickpoints show that the difference between these parameters derived in each DEM is subtle, with only the $k_{sn}$ and the locations of the knickpoints differing.

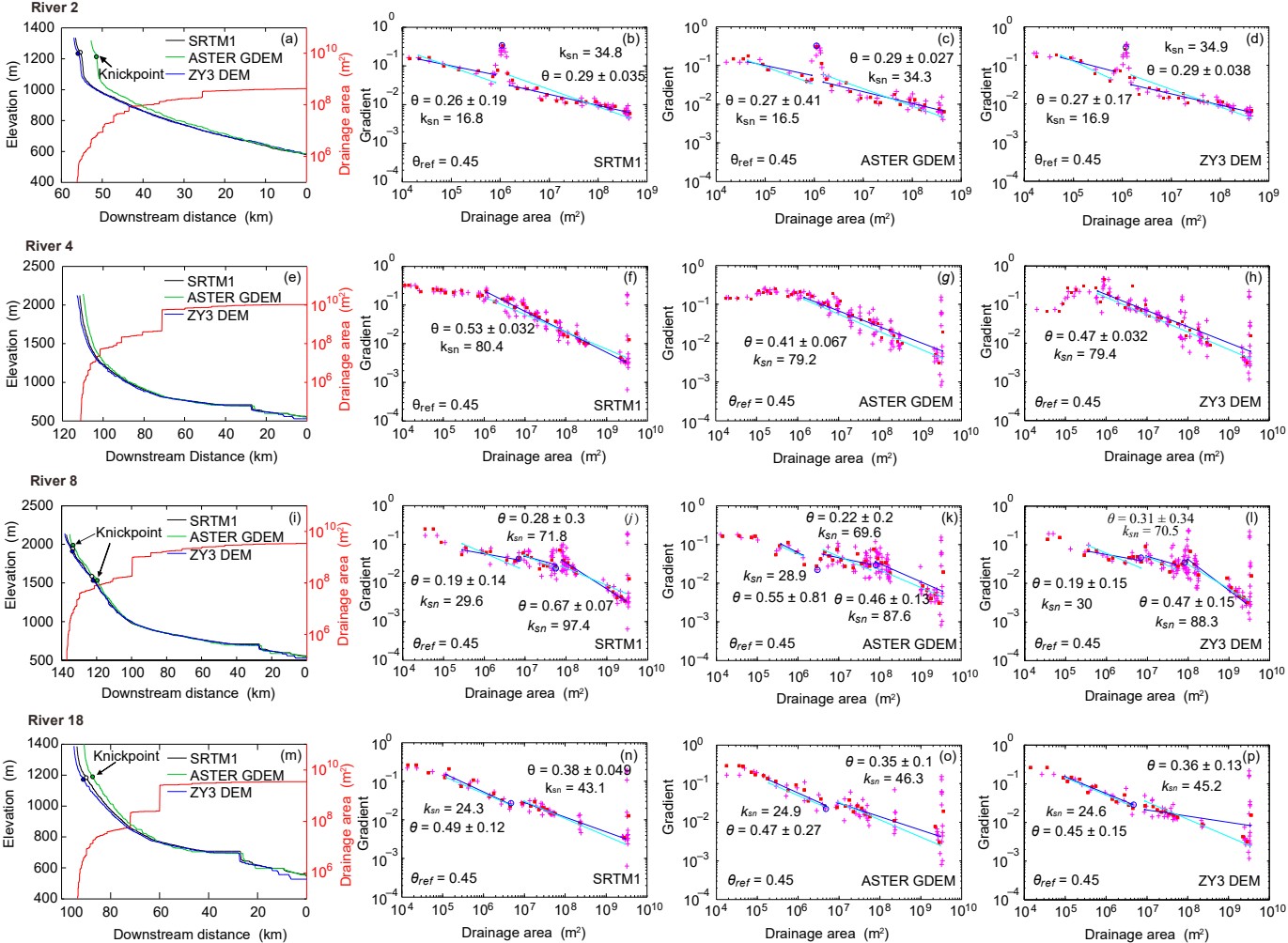

**Figure 9.** Representative examples of river longitudinal profiles, and SA plot (log-log SA, where the relationship is between local channel gradient and drainage area). $K_{sn}$ was derived from SRTM, ASTER GDEM, and ZY3 DEM by using a reference concavity $\theta_{ref} = 0.45$. Rivers 2, 4, 8, and 18 are the rivers with one vertical-step knickpoint, no knickpoint, two slope-break knickpoints, and one slope-break knickpoint. The red lines in the first column are the drainage area of upstream. (**a**,**e**,**i**,**m**) are the longitudinal profiles and drainage areas along the downstream distance for River 2, 4, 8, and 18, respectively. (**b**,**f**,**j**,**n**) are the knickpoint extraction from the SRTM1 for River 2, 4, 8, and 18, respectively. (**c**,**g**,**k**,**o**) are the knickpoint extraction from the ASTER GDEM for River 2, 4, 8, and 18, respectively. (**d**,**h**,**l**,**p**) are the knickpoint extraction from the ZY3 DEM for River 2, 4, 8, and 18, respectively.

Firstly, the length and shape of the ZY3 DEM are the same as those in SRTM1, while the river profiles in ASTER DEM are slightly shorter than they are in the other DEMs (Figure 9 and Table 5). Taking River 4 as an example, the length of the river profiles in these DEMs (SRTM1, ASTER GDEM, and ZY3 DEM) are 91.9, 86.9 and 87.4 km, respectively. Furthermore, the SRTM1 river long profiles are smoother, while the ASTER GDEM and ZY3 DEM profiles demonstrate some shaking (Figure 9a,e,i,m), suggesting that there is more noise in ASTER GDEM and ZY3 DEM than in SRTM1 and that the SRTM1 profile is more consistent with the true long profiles. The noise of the SA plots in SRTM1 and ZY3 DEM is lower than that in ASTER GDEM, indicating that smoothing is necessary before $k_{sn}$ can be extracted in ASTER GDEM.

**Table 5.** Comparison of river long profiles related to different DEMs. * donates rivers with second (higher) knickpoints.

| River No. | DEMs | Knickpoint Elevation | Total Drainage Area | River Length | Upstream Distance of Knickpoint | $k_{sn}$ above Knickpoint | θ | ± | $k_{sn}$ below Knickpoint | θ | ± | Knickpoint Retreat Rates (at 1.4 Myr) |
|---|---|---|---|---|---|---|---|---|---|---|---|---|
| | | (m) | (km$^2$) | (km) | (km) | (m$^{0.9}$) | | | (m$^{0.9}$) | | | (mm/Year) |
| 2 | SRTM1 | 1219 | 3.3 | 56.3 | 0.4 | 16.8 | 0.26 | 0.19 | 34.8 | 0.29 | 0.04 | 0.3 |
| | ASTER GDEM | 1205 | 3.1 | 52.4 | 0.3 | 16.5 | 0.27 | 0.41 | 34.3 | 0.29 | 0.03 | 0.2 |
| | ZY3 DEM | 1213 | 3.2 | 56.6 | 0.4 | 16.9 | 0.27 | 0.17 | 34.9 | 0.29 | 0.04 | 0.3 |
| 4 | SRTM1 | – | 91.9 | 111.4 | – | 80.4 | 0.53 | 0.03 | – | – | – | – |
| | ASTER GDEM | – | 86.9 | 108.7 | – | 79.2 | 0.41 | 0.07 | – | – | – | – |
| | ZY3 DEM | – | 87.4 | 111.7 | – | 79.4 | 0.47 | 0.03 | – | – | – | – |
| 8 * | SRTM1 | 1909 | 115.8 | 137.2 | 26.8 | 29.6 | 0.19 | 0.14 | 71.8 | 0.28 | 0.30 | 19.1 |
| | | 1562 | – | – | 15.4 | 71.8 | 0.28 | 0.30 | 97.4 | 0.67 | 0.07 | 11.0 |
| | ASTER GDEM | 1973 | 112.1 | 134.6 | 28.3 | 28.9 | 0.55 | 0.81 | 69.6 | 0.22 | 0.20 | 20.2 |
| | | 1537 | – | – | 14.4 | 69.6 | 0.22 | 0.20 | 87.6 | 0.46 | 0.13 | 10.3 |
| | ZY3 DEM | 1894 | 113.4 | 137.6 | 28.9 | 30.0 | 0.19 | 0.15 | 70.5 | 0.31 | 0.34 | 20.6 |
| | | 1541 | – | – | 14.8 | 70.5 | 0.31 | 0.34 | 88.3 | 0.47 | 0.15 | 10.6 |
| 18 | SRTM1 | 1184 | 67.7 | 93.2 | 21.2 | 24.3 | 0.49 | 0.12 | 43.1 | 0.38 | 0.05 | 15.1 |
| | ASTER GDEM | 1195 | 63.3 | 90.4 | 20.5 | 24.9 | 0.47 | 0.27 | 46.3 | 0.35 | 0.10 | 14.6 |
| | ZY3 DEM | 1172 | 64.9 | 94.6 | 21.6 | 24.6 | 0.45 | 0.15 | 45.2 | 0.36 | 0.13 | 15.4 |

Secondly, there is no obvious change in the $k_{sn}$ above and below the knickpoint in River 2 (Figure 9), where the $k_{sn}$ ratios for SRTM1, ASTER GDEM, and ZY3 DEM are 2.07, 2.08 and 2.07, respectively, indicating that the knickpoint in River 2 has no tectonic significance. The rivers with one or two slope-break knickpoints derived from ZY3 DEM have a slightly increased $k_{sn}$ as compared with SRTM1 (Table 5), while the ratios of $k_{sn}$ above the knickpoints are similar. The value of $k_{sn}$ is lower in ASTER GDEM than the other DEM datasets. This is particularly apparent in River 4 without the knickpoint, for which the $k_{sn}$ of the ASTER GDEM river profile is 79.2 $m^{0.9}$, while that of SRTM1 is 80.4 $m^{0.9}$. Generally, the three DEMs still indicate that the $k_{sn}$ above the knickpoint is lower than the $k_{sn}$ below the knickpoint.

Finally, the concavity is similar in all the DEMs, with a medium range of $\theta = 0.19–0.67$, with a lower northern margin, where $\theta = 0.35–0.49$. The range of the concavity index for the ASTER GDEM is more dispersed, while that of the ZY3 DEM is generally aggregated. Overall, the concave index is more variable but does not show obvious changes.

*4.3. River Incision Extraction*

Previous studies have indicated that terraces are composed of alluvial deposits [35], with the ages provided by electron spin resonance (ESR) and optical simulating luminescence (OSL) for each terrace in the Qianhe Basin (T1 to T5) at 41, 127, 375–505, 788 and 1411 kyr, respectively [88]. Therefore, six profiles through the river terraces were used to extract heights of the river terraces, and the main results were shown in Table 6.

**Table 6.** The elevation of the terrace derived from SRTM1, ASTER GDEM, and ZY3 DEM. The periods of the terraces collected from [88].

| Profiles | Terrace | Height of Front Edge (m) | | | Height of Rear Edge (m) | | | Height of Terrace (m) | | | Period | Incision Rates (m/kyr) | | |
|---|---|---|---|---|---|---|---|---|---|---|---|---|---|---|
| | | SRTM1 | ASTER | ZY3 | SRTM1 | ASTER | ZY3 | SRTM1 | ASTER | ZY3 | (Kyr) | SRTM1 | ASTER | ZY3 |
| AA′ | T1 | 5.3 | 7.1 | 3.9 | 12.5 | 9.8 | 12.4 | 8.9 | 8.5 | 8.2 | 41 | 0.21 | 0.21 | 0.20 |
| | T3 | 95.3 | 129.1 | 97.9 | 112.4 | 130.4 | 108.1 | 103.9 | 129.8 | 103.0 | 505 | 0.21 | 0.26 | 0.20 |
| | T4 | 131.3 | 141.6 | 130.3 | 135.7 | 160.1 | 136.4 | 133.5 | 150.9 | 133.4 | 788 | 0.17 | 0.19 | 0.17 |
| BB′ | T1 | 3.3 | 4.6 | 3.7 | 7.2 | 10.3 | 8.1 | 5.3 | 7.5 | 5.9 | 41 | 0.13 | 0.18 | 0.14 |
| | T2 | 54.8 | 52.3 | 53.8 | 58.3 | 62.6 | 61.0 | 56.6 | 57.5 | 57.4 | 127 | 0.45 | 0.45 | 0.45 |
| | T3 | 112.2 | 113.3 | 105.5 | 122.0 | 139.0 | 119.1 | 117.1 | 125.6 | 112.3 | 505 | 0.23 | 0.25 | 0.22 |
| | T4 | 142.7 | 145.0 | 145.0 | 152.6 | 147.3 | 159.9 | 147.7 | 146.2 | 152.5 | 788 | 0.19 | 0.19 | 0.19 |
| | T5 | 162.7 | 169.7 | 159.7 | 183.0 | 183.3 | 185.9 | 172.9 | 176.5 | 172.8 | 1411 | 0.12 | 0.13 | 0.12 |
| CC′ | T1 | 3.6 | 12.6 | 5.5 | 17.4 | 20.3 | 18.8 | 10.5 | 16.5 | 13.1 | 41 | 0.26 | 0.40 | 0.30 |
| | T2 | 47.7 | 52.8 | 48.3 | 67.1 | 55.3 | 67.1 | 57.4 | 54.1 | 57.7 | 127 | 0.45 | 0.43 | 0.45 |
| | T3 | 100.6 | 106.2 | 97.3 | 112.2 | 120.0 | 115.7 | 106.4 | 113.1 | 106.5 | 505 | 0.21 | 0.22 | 0.21 |
| | T4 | 156.9 | 153.1 | 163.3 | 172.9 | 173.2 | 168.1 | 164.7 | 163.2 | 165.7 | 788 | 0.21 | 0.21 | 0.21 |
| DD′ | T1 | 4.6 | 8.7 | 6.4 | 9.9 | 13.2 | 10.0 | 7.3 | 11.0 | 8.2 | 41 | 0.18 | 0.27 | 0.20 |
| | T2 | 32.5 | 44.1 | 37.4 | 38.9 | 55.2 | 43.7 | 35.7 | 49.7 | 40.6 | 127 | 0.28 | 0.39 | 0.32 |
| | T3 | 74.8 | 66.7 | 79.0 | 86.1 | 96.9 | 96.4 | 80.5 | 81.8 | 87.7 | 375 | 0.21 | 0.22 | 0.23 |
| | T4 | 93.9 | 102.1 | 93.6 | 109.7 | 125.2 | 117.7 | 101.8 | 113.7 | 105.7 | 788 | 0.13 | 0.14 | 0.13 |
| EE′ | T1 | 3.1 | 7.5 | 2.7 | 7.0 | 8.8 | 9.0 | 5.1 | 8.2 | 5.9 | 41 | 0.12 | 0.20 | 0.14 |
| | T2 | 7.8 | 25.7 | 10.1 | 12.7 | 36.4 | 16.0 | 10.3 | 31.1 | 13.1 | 127 | 0.08 | 0.24 | 0.10 |
| | T3 | 52.2 | 90.8 | 52.7 | 71.7 | 110.3 | 77.4 | 62.0 | 100.6 | 65.1 | 375 | 0.17 | 0.27 | 0.17 |
| | T4 | 127.2 | 133.5 | 122.9 | 157.7 | 157.3 | 160.4 | 142.5 | 145.4 | 141.7 | 788 | 0.18 | 0.18 | 0.18 |
| FF′ | T1 | 10.7 | 11.2 | 10.0 | 16.2 | 19.2 | 17.2 | 13.5 | 15.2 | 13.6 | 41 | 0.33 | 0.37 | 0.33 |
| | T2 | 29.6 | 36.3 | 33.5 | 39.0 | 52.6 | 44.7 | 34.3 | 44.5 | 39.1 | 127 | 0.27 | 0.35 | 0.31 |
| | T3 | 84.3 | 81.2 | 85.5 | 103.5 | 106.2 | 103.1 | 93.9 | 93.7 | 94.3 | 375 | 0.25 | 0.25 | 0.25 |
| | T4 | 126.2 | 126.5 | 129.8 | 166.8 | 162.2 | 172.0 | 146.5 | 144.4 | 150.9 | 788 | 0.19 | 0.18 | 0.19 |

As shown in Figure 10, the river terraces (T) are asymmetrically distributed on both sides of the Qianhe River. T2 and T3 on the northern margin have eroded rapidly, while the four terraces on the southern margin are symmetrical in location. The terrace height was extracted from the height difference between the front and rear edges of the terrace (Table 6). The average northern terrace heights (T1, T2, T3, T4 and T5) measured from SRTM1 are 8.2, 57.0, 109.1, 148.6 and 172.9 m, respectively, while the southern terraces height (T1, T2, T3 and T4) are 8.6, 26.7, 78.8, and 130.3 m, respectively. For ZY3 DEM, the heights for northern terrace are 1.1, 1.0, 1.0, 1.0 and 1.0 times higher than the heights from

SRTM1, while the southern terrace are 1.1, 1.2, 1.0 and 1.0 times higher than the SRTM1. The heights on the northern terraces obtained from ASTER GDEM are not much different from the other two DEMs, more specifically, the terrace heights are 1.3, 1.0, 1.1, 1.0 and 1.0 times higher than SRTM1, respectively. However, the southern terrace heights extracted from ASTER GDEM are 1.3, 1.6, 1.2 and 1.0 times higher than the SRTM1, indicating that the southern terraces are experiencing a rapid erosion. Additionally, the river terrace heights extracted from the DEMs are consistent with the previous studies [35], indicating that these results can be used to investigate the river incision.

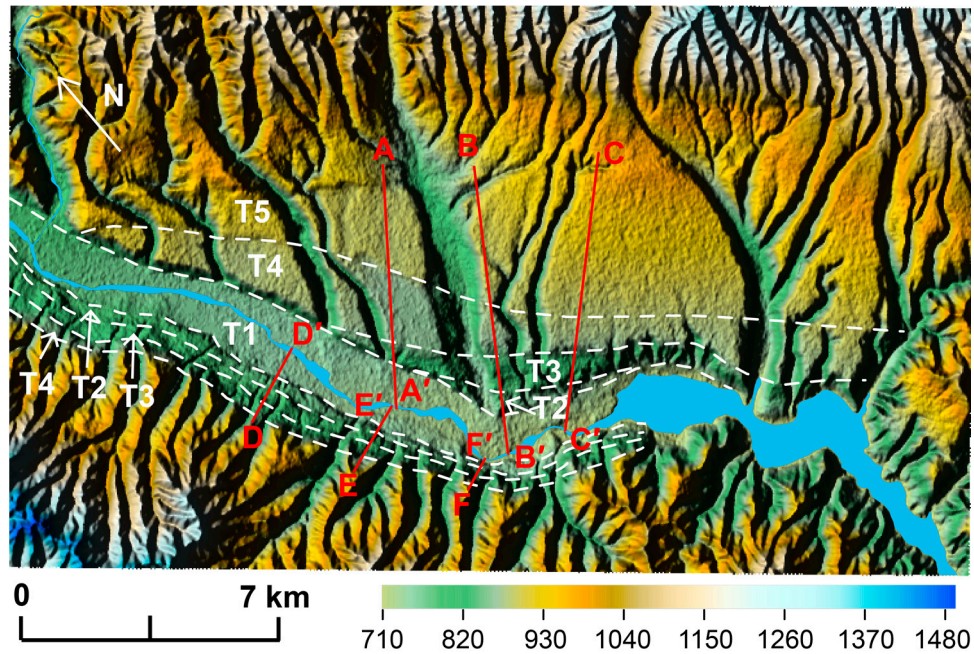

**Figure 10.** The river terraces in Qian Basin. The white dotted lines are the boundaries of terraces in the Qianhe Basin, modified from [35,74]. The red lines (AA′, BB′, CC′, DD′, EE′, and FF′) are the topographic profiles of the terrace height.

Furthermore, we estimated the river incision rates in SRTM1, ASTER GDEM, and ZY3 DEM and found the main differences in ASTER GDEM, with more discrepancies in elevation than the other datasets. More specifically, due to the higher precision of SRTM1 (Tables 3 and 4), the heights in SRTM1 can be regarded as reference data, and the northern river incision rates are estimated to be in the range 0.12–0.45 m/kyr, while the southern rates are in the range 0.08–0.32 mm/kyr. As for the ZY3 DEM, the river incision rates for the northern and southern rates are 0.12–0.45 and 0.10–0.33 mm/year, respectively. Thus, the northern rivers have higher incision rates. The RMSE of the incision rates in ASTER DEM and ZY3 DEM are 0.06 and 0.02 m/kyr, respectively, indicating that ASTER GDEM is inferior to ZY3 DEM.

## 5. Discussion

### 5.1. Which DEM Is Better for Investigating the River Long Profiles?

Previous studies have shown that resolution has a minimal effect on the final river network [89], suggesting that the differences in the resolutions of these DEMs would not differ in the final long profiles, but would show more noise in the rough channel profiles. However, the presence of noise in the SA plot for the ASTER GDEM indicates that ASTER GDEM is inappropriate for knickpoint extraction without smoothing. Furthermore, the elevation difference between the ASTER GDEM and the measured DEM is greater in areas with higher relief and slope [90], which will undoubtedly affect the length and shape of the final river long profiles. Additionally, sink-free (bulge in the valley) DEMs were recognized as sink and fill, resulting in more abnormalities and noise in the SA plot of ASTER GDEM

than in the SRTM1 and ZY3 DEM datasets (Figure 9c,g,k,o), with the largest difference in elevation (Tables 3 and 4). As a result, the ASTER GDEM was considered to be inferior in river profile analysis [91] and scattering channel slope [92], with SRTM1 demonstrating a smoother channel profile and better quality. Furthermore, the retreat rate (the propagation rate of the knickpoint from the faults) is also new evidence for changes in the tectonic uplift [10,73,93], where the southern retreat rates of the knickpoints are larger than the northern rates that correspond to the fault initiation period of 1.2–1.4 Myr [9,35]. The retreat rates of the knickpoints in ASTER GDEM are the lowest with a range of 0.2–20.2 mm/year in the representative tributaries (Table 5), while the ZY3 DEM is the highest with a range of 0.3–20.6 mm/year. The distribution of the retreat rates indicates that higher uplift occurred in the southern margin of the Qianhe Basin than in the northern margin. Thus, the lower retreat rates of the knickpoints will result in the lower uplift rates indicated by the presence of voids in ASTER GDEM, while the ZY3 DEM is more consistent with the previous results and can obtain the change in rate over a long time period.

Notably, previous studies have shown that SRTM can penetrate the vegetation canopy [94], suggesting that the SRTM1 could observe the elevation of the real surface with more accurate, which would be significantly better than ASTER GDEM. However, there is no evidence indicating that the SRTM1 data is accurate as the RMSE of 9.04 m is linked to the 1:50,000 DEM (Table 4). Moreover, SRTM1 lacks detailed information, especially in some mountainous and valley areas. As a novel representative data of stereo images and a method of generating DEMs, ZY3 DEM has a similar elevation difference to SRTM1, resulting in approximate knickpoints and $k_{sn}$, indicating that ZY3 DEM could be a replacement dataset for river long profiles. In particular, the continuous observation of active tectonics suggests that the DEM generated from the ZY3 stereo images is superior to SRTM1. However, the quality and distribution of the GCPs affect the final accuracy of ZY3 DEM, even though the precision of the GCPs is better than 1 m.

### 5.2. How to Evaluate the Vertical Accuracy in Measuring the River Incision?

Generally, several factors limit the extraction of river incision rates from ZY3 DEM, such as the distribution and precision of GCPs, the mosaic algorithm of the stereo pairs, the RFM model, topography, and the gradient of inclines in an area [56,95–97]. More specifically, a large systematic error exists in the positioning accuracy of the direct forward intersection when RFM parameters are used, but this could be compensated for by using the constraint relationships among the images [55], indicating that more GCPs should be selected to improve the relationship between the stereo pairs. Tang et al. [98] suggested that the vertical accuracy would improve when more GCPs were applied. However, the distribution and precision of the GCPs are also determined by the regional geomorphology (e.g., gradient, hill shadow, and position) and no more points could be chosen, especially in the Qianhe Basin where the mountainous areas are induced by active faulting. Furthermore, the GPS measurement is limited to the vertical coverage of vegetation, resulting in an imprecise absolute orientation.

The uplift of the Liupan–Longxi and Ordos blocks has a significant effect on river erosion in the Qianhe Basin. Gao et al. [99] reported that the river incision of the upper Weihe River in the Longxi Basin ranges from 0.09 to 0.32 m/kyr. Moreover, they also measured the age and height of the terraces running alongside the Upper Weihe River in the Sanyangchuan Basin, and found that the incision rates ranged from 0.21 to 1.03 m/kyr [100]. Zhao et al. [101] argued that the incision rates in the upper Liupanshan mountains, located at the northern margin of the Qianhe Basin and the southwest margin of the Ordos Block, range from 0.37 to 1.13 m/kyr. All of these results indicate a decreasing trend in the rate of incision from the margin to the central area, obeying the characteristic of the "high-low-high" pattern discussed by Gao et al. [100] and the changes in the topography and the gradients of inclines in the area. Moreover, the changes in landform have been confirmed unlikely to have been caused by the climate [9], but may limit the erosion rates, rendering the rate of erosion lower than that of uplift. This indicates that the uplift of the eastward

extrusion Liupan–Longxi mountains may also increase the downcutting of the rivers. This new finding not only provides some guidance for the spatial distribution of the tectonic deformation but also stresses that the river terraces in the area result from the gradual erosion of rivers under changes in tectonic uplift that have been taking place since the late Cenozoic.

## 6. Conclusions

In this study, the data from China's Ziyuan-3 satellite stereo images were introduced for generating DEMs that can be used to delineate the fluvial index and river incision rate in the Qianhe Basin. To evaluate the vertical accuracy, ICESat/GLAH14 point cloud and regional 1:50,000 high-precision DEMs were used to compare the accuracy of ZY3 DEM with two global free datasets (SRTM1 and ASTER GDEM). The RMSE of the elevation difference in the DEM generated by the ZY3 satellite stereo images is approximately 9.31 and 9.71 m, which is comparable with GLAH14 and 1:50,000 DEM and is in good agreement with SRTM1. Analysis of the longitudinal profiles in the Qianhe Basin produced similar knickpoints, lengths, and values for $k_{sn}$ from SRTM1 and ZY3 DEM, while ASTER GDEM had the lowest characteristics in terms of river long profiles. The river incision rates were also estimated using the ages and heights of the river terraces in central Qianhe Basin, with the northern terraces ranging from 0.12 to 0.45 m/kyr, and the southern terraces ranging from 0.10 to 0.33 m/kyr.

Using this method, changes in the river long profiles and terraces can be obtained from the ZY3 stereo images, and a more accurate response to active faulting can be found, while the other two datasets are expensive and difficult to use. However, the SRTM1 data have more accurate profiles, which is consistent with the results of previous studies, although the ZY3 stereo images can provide a higher resolution of <2 m. Notably, the good agreement between ZY3 DEM and SRTM1 suggests that the former DEM could be a better alternative for further work in tectonic geomorphology. Because of the sinks in ASTER GDEM, particular attention should be paid to the filling algorithms before using these data.

However, several shortcomings should be taken into account when extracting river long profiles and terrace heights by using the DEM generated from stereo images as follows: (1) Owing to the limitation of the terrain changes and the upper vegetation, incorrect DEMs may have resulted from the GCP measurement, meaning that a more accurate absolute orientation cannot be provided using this method, especially in the mountainous area of the Qianhe Basin that has been induced by higher tectonic uplift. (2) The dislocation in the two stereo pairs may increase the number of knickpoints in the long profiles, stressing that more attention should be paid to the relationship between the two stereo pairs when mosaicking the DEMs. (3) The extraction of river longitudinal profiles and incision rates, although it can be done with the help of computers, still relies heavily on the knowledge background of the researcher, the production and quality of the DEM, and the running rate of the code program. In fact, only exploring the longitudinal profile of the river geomorphology is not enough as the cross-sectional variation can also respond to tectonic changes (e.g., river width generated from the DEM), and the differences of DEM in this area should also be expanded in the future.

**Author Contributions:** Conceptualization, Z.L. and L.H.; methodology, Z.L., L.H., Z.Y., H.C., F.G. and J.G.; software, Z.L., Z.Y., H.C., F.G., J.G. and Y.J.; formal analysis, Z.L. and L.H.; investigation, Z.L., Z.Y, H.C., F.G., J.G. and Y.J.; data curation, Z.L., Z.Y., H.C., F.G., J.G. and Y.J.; writing—original draft preparation, Z.L.; writing—review and editing, Z.L.; visualization, Z.L.; project administration, L.H.; funding acquisition, L.H. All authors have read and agreed to the published version of the manuscript.

**Funding:** This research was funded by the "Design and construction technology of gully-slope treatment project based on ecological safety", grant number 2017YFC0504705-02, and the Shaanxi Key Laboratory of Land Consolidation Open Fund Program, grant number 2018-ZY01.

**Institutional Review Board Statement:** Not applicable.

**Informed Consent Statement:** Not applicable.

**Data Availability Statement:** The data presented in this study are available on request from the corresponding author. The data are not publicly available due to collecting data by the team and the partner, the partner demanded strictly not sharing data.

**Conflicts of Interest:** The authors declare no conflict of interest.

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
