# Peer review of "Evaluating the Vertical Accuracy of DEM Generated from ZiYuan-3 Stereo Images in Understanding the Tectonic Morphology of the Qianhe Basin, China"

_remotesensing, doi:10.3390/rs13061203_

Round 1
Reviewer 1 Report
Dear Author,
Article is in good quality, but need some changes to make it clear.
1 In Figure 1 can understand coloured lines with number is it elevation if yes is it in meters.
2 Why used old data and information Example satellite data from 2013.
3 Will be better if in 2.2.2 will write widely about generation of different resolution data and accuracy check.
4 To much tables without deep explanation. For example Table 6.
5 Is this paper has comparison analysis of field and satellite results?
Reviewer 2 Report
Although the paper results are really interesting, the overall presentation is confusing. You paper aim to two different things:
1 - which is the "best" DEM for morphological analysis
2 - what are the uplift and erosional signals of the study area
The task 1 is poorly supported by your paper whereas the second aim is much more sounding.
I suggest to revisit the paper, simplifying the firs part and deepening the second part.
Reviewer 3 Report
This paper compares the vertical accuracy of a DEM generated from ZiYuan satellite imagery to that of the widely available SRTM and ASTER products for use in extracting river longitudinal profiles in a region of tectonic uplift in China. Using satellite remote sensing to infer rates of tectonic uplift and river incision based on the age and height of terraces is a useful application, because field-based studies can be difficult to conduct in remote mountainous regions. The research methods are sound and the analysis and conclusions are well-supported. I have suggestions for three areas in which the paper could be improved: 1) There are numerous minor problems with the English writing throughout the paper, which mostly do not interfere with the reader’s ability to understand the paper, but which could be polished to produce a higher-quality publication. 2) The usefulness of the ZY3 satellite data and the generalizability of the findings should be better justified. 3) Results and discussion should be more clearly separated. Please see specific comments below.
Title: The current version of the title exemplifies the English language errors that are common throughout the paper, most of which are related to the use of prepositions or articles. There needs to be a preposition between “Evaluation” and “the” in the title: “Evaluation of the Vertical Accuracy…” Also, there needs to be an article before “DEM”: “Evaluation of the Vertical Accuracy of a DEM…” Finally, capitalization should be applied consistently. Since the rest of the title is in title case, “Vertical Accuracy” should also be capitalized. There are lots of errors like this throughout the paper, but I wanted to highlight these ones because they are especially prominent when they appear in the title.
Section 2.2.1: It’s not clear to me what value ZY3 adds beyond that of a dataset like SRTM. I understand that ZY3 has a higher temporal resolution than SRTM or ASTER, but why is temporal resolution important for this particular application? You’re looking at long-term landscape evolution, so frequent revisit times are not needed. If you were instead monitoring seismic hazards in real-time, I could see why the temporal resolution would be important, but those kinds of movements are probably not detectable within the vertical accuracy of any of these datasets. What is not emphasized in the text here is that ZY3 has a much higher spatial resolution than SRTM or ASTER, which seems like it would be a more important advantage for this application given that the smaller pixel size could potentially improve the vertical accuracy. So there needs to be a clearer justification of the advantages that ZY3 offers over existing datasets such as SRTM in estimating river incision rates. Otherwise, the paper demonstrates that ZY3 provides similar accuracy to SRTM but does not explain why you wouldn’t just use SRTM instead. This is especially the case because SRTM data is free available but it seems that ZY3 data is not (some clarification on the availability of ZY3 data would also be helpful here so that readers know whether this is a dataset they can use or not).
Figure 3 and Figure 8: There should be scale bars in these maps.
Figure 7, Table 5, Figure 8, Table 6: These are results, so they should be presented in the Results section rather than the Discussion section. The text relating to the interpretation and significance of these results can remain in the Discussion section.
Page 17, lines 10-11: I believe there are some typos here: “the southern rates are 0.08. 0.33 mm/kyr” should be “the southern rates are 0.08-0.33 m/kyr”. Although there is also a slight inconsistency in that, in the abstract and conclusion, the lower limit of incision for the southern terraces is given as 0.10 m/kyr instead of 0.08 m/kyr.
Conclusions: The results and conclusions of this study seem very specific to the study area and the ZY3 data. The last paragraph of the paper is helpful in drawing some lessons about the methods of extracting longitudinal profiles and terraces heights in a DEM generated from satellite stereo images. It would be helpful to expand upon these more generalized findings because they are likely to be more applicable to other study regions and/or datasets.
Round 2
Reviewer 2 Report
No other suggestions